# Merge-and-Shrink Task Reformulation for Classical Planning

**Álvaro Torralba**
Saarland University, Saarland Informatics Campus
Saarbrücken, Germany
torralba@cs.uni-saarland.de

**Silvan Sievers**
Basel University
Switzerland
silvan.sievers@unibas.ch

## Abstract

The performance of domain-independent planning systems heavily depends on how the planning task has been modeled. This makes task reformulation an important tool to get rid of unnecessary complexity and increase the robustness of planners with respect to the model chosen by the user. In this paper, we represent tasks as factored transition systems (FTS), and use the merge-and-shrink (M&S) framework for task reformulation for optimal and satisfying planning. We prove that the flexibility of the underlying representation makes the M&S reformulation methods more powerful than the counterparts based on the more popular finite-domain representation. We adapt delete-relaxation and M&S heuristics to work on the FTS representation and evaluate the impact of our reformulation.

## Introduction

Classical planning deals with the problem of finding a sequence of actions that achieve a set of goals, given a model of the world that describes an initial state and a set of available actions. For representing the problem, different planning formalisms can be used, the most common being STRIPS or finite-domain representation (FDR). The choice of formalism does not change the complexity of the problem, which is PSPACE-complete (Bylander 1994; Bäckström and Nebel 1995). However, it may impact the so-called accidental complexity, when the structure of the task is disguised by how it is encoded (Haslum 2007). Accidental complexity can be dealt with by reformulating the planning task prior to solving it. There are several reformulation methods based on, e.g., downward-refinable abstractions (Haslum 2007) or tunnel macros (Coles and Coles 2010), which can be combined to reduce the size of FDR tasks (Tozicka et al. 2016).

Merge-and-Shrink (M&S) is a general framework to generate abstractions, originally defined in the model-checking area (Dräger, Finkbeiner, and Podelski 2006; 2009), that can be used to derive an admissible heuristic (Helmert, Haslum, and Hoffmann 2007; Helmert et al. 2014) and/or detect unsolvability (Hoffmann, Kissmann, and Torralba 2014). Further work on the topic noticed that this can be understood as applying transformations to a set of transition systems (Sievers, Wehrle, and Helmert 2014) and hence as a method to transform planning tasks in the factored transition system

(FTS) representation (Torralba and Kissmann 2015). However, these methods perform the search on an FDR task, only using M&S to derive heuristics or remove irrelevant actions.

In this paper, we use M&S as a task reformulation method on FTS tasks. We show that some of the M&S transformations originally devised for constructing abstraction heuristics can also be used for optimal and satisfying reformulation. To do so, we provide algorithms that transform solutions for the reformulated task into plans for the original task. We also show that our M&S reformulations dominate their counterparts based on FDR representations, i.e., a suitable combination of existing M&S transformations can always do the same (and sometimes more) simplifications to any task.

To search on the FTS representation, planning algorithms and heuristics originally devised for STRIPS or FDR tasks must be adapted. As the FTS formalism is slightly more expressive than FDR, this is similar to adapting algorithms to support (a limited form of) disjunctive preconditions and conditional effects. We adapt heuristic search methods with M&S and delete-relaxation heuristics for the FTS representation. Our experimental study shows the potential of these reformulations to reduce the state space and speed-up the search. Full proofs and additional experimental results are included in a technical report (Torralba and Sievers 2019).

## Representation of Planning Tasks

A planning task is a compact representation of a TS. A *transition system* (TS) is a tuple $\Theta = \langle S, L, T, s^I, \mathcal{S}^\star \rangle$ where $S$ is a finite set of *states*, $L$ is a finite set of *labels* each associated with a *label cost* $c(\ell) \in \mathbb{R}_0^+$, $T \subseteq S \times L \times S$ is a set of *transitions*, $s^I \in S$ is the *initial state*, and $\mathcal{S}^\star \subseteq S$ is the set of *goal states*. We use $s \in \Theta$ to refer to states in $\Theta$ and $s \xrightarrow{\ell} t \in \Theta$ to refer to its transitions. An *s-plan* for a state $s$ is a path from $s$ to any $s^* \in \mathcal{S}^\star$. Its cost is the summed label costs of all labels of the path. The *perfect heuristic*, $h^*(s)$, is the cost of a cheapest $s$-plan. An $s$-plan is *optimal* iff its cost equals $h^*(s)$. A plan for $\Pi$ is an $s^I$-plan.

An abstraction is a function $\alpha$ mapping states in $\Theta$ to a set of *abstract states* $S^\alpha$. The *abstract state space* $\Theta^\alpha$ is $\langle S^\alpha, L, T^\alpha, s_\alpha^I, \mathcal{S}_\alpha^\star \rangle$, where $\alpha(s) \xrightarrow{\ell} \alpha(s') \in T^\alpha$ iff $s \xrightarrow{\ell} s'$ in $\Theta$, $s_\alpha^I = \alpha(s^I)$, and $\mathcal{S}_\alpha^\star = \{\alpha(s) \mid s \in \mathcal{S}^\star\}$.

An *FDR task* is a tuple $\Pi^\mathcal{V} = \langle \mathcal{V}, \mathcal{A}, s^\mathcal{I}, \mathcal{G} \rangle$. $\mathcal{V}$ is a finite set of *variables* $v$, each with a *finite domain* $D_v$. A *partial*

*state* is a function $s$ on a subset $\mathcal{V}(s)$ of $\mathcal{V}$, so that $s(v) \in D_v$ for all $v \in \mathcal{V}(s)$; $s$ is a *state* if $\mathcal{V}(s) = \mathcal{V}$. $s^{\mathcal{I}}$ is the *initial state* and the *goal* $\mathcal{G}$ is a partial state. $\mathcal{A}$ is a finite set of *actions*. Each $a \in \mathcal{A}$ is a tuple $\langle pre_a, e\!f\!f_a, c(a) \rangle$ where $pre_a$ and $e\!f\!f_a$ are partial states, called its *precondition* and *effect*, and $c(a) \in \mathbb{R}_0^+$ is its *cost*. An action $a$ is applicable in a state $s$ if $\forall_{v \in \mathcal{V}(pre_a)} s(v) = pre_a(v)$. Applying it yields the successor state $s[\![a]\!]$ with $s[\![a]\!](v) = e\!f\!f_a(v)$ if $v \in \mathcal{V}(e\!f\!f_a)$ and $s[\![a]\!](v) = s(v)$ otherwise.

The state space of an FDR task $\Pi^{\mathcal{V}}$ is a TS $\Theta = \langle S, L, T, s^I, \mathcal{S}^\star \rangle$ where $S$ is the set of all states, $s^I = s^{\mathcal{I}}$, $s \in \mathcal{S}^\star$ iff $\forall_{v \in \mathcal{V}(\mathcal{G})} \mathcal{G}(v) = s(v)$, $L = \mathcal{A}$, and $s \xrightarrow{a} s[\![a]\!] \in T$ if $a$ is applicable in $s$.

An *FTS task* is a set of TSs $\{\Theta_1, \ldots, \Theta_n\}$ with a common set $L$ of labels. The synchronized product $\Theta_1 \otimes \Theta_2$ of two TSs is another TS with states $S = \{(s_1, s_2) \mid s_1 \in \Theta_1 \wedge s_2 \in \Theta_2\}$, labels $L = L_1 = L_2$, transitions $T = \{(s_1, s_2) \xrightarrow{\ell} (s_1' s_2') \mid s_1 \xrightarrow{\ell} s_1' \in \Theta_1 \wedge s_2 \xrightarrow{\ell} s_2' \in \Theta_2\}$, initial state $s^{\mathcal{I}} = (s_1^{\mathcal{I}}, s_2^{\mathcal{I}})$, and goal states $\mathcal{S}^\star = \{(s_1, s_2) \mid s_1 \in \mathcal{S}_1^\star \wedge s_2 \in \mathcal{S}_2^\star\}$.

The state space of an FTS task $\Pi^{\mathcal{T}} = \{\Theta_1, \ldots, \Theta_n\}$ is defined as $\Theta = \Theta_1 \otimes \cdots \otimes \Theta_k$. Whenever it is not clear from context, we will use subscripts to differentiate states in the state space ($s, s', t \in \Theta$) and in the individual components ($s_i, s_i', t_i \in \Theta_i$). Given $s \in \Theta$, we write $s[\Theta_i]$ to refer to the projection of $s$ onto $\Theta_i$. A solution $\pi$ for an FTS task is a sequence $s_0 \xrightarrow{\ell_1} s_1 \xrightarrow{\ell_2} \ldots \xrightarrow{\ell_k} s_k$ such that $s_k \in \mathcal{S}^\star$.

There is a close connection between FTS and FDR tasks, since TSs in an FTS task correspond to FDR variables with domain equal to the set of states of the TS. Then, states in FDR (which are assignments of values to variables) correspond to states in the FTS representation, which are an assignment of states $s_i$ to each $\Theta_i$. Given an FDR task $\Pi^{\mathcal{V}}$ it is simple to construct the corresponding FTS task, which we call the *atomic* representation of $\Pi^{\mathcal{V}}$. There is a TS $\Theta_v$ for every variable $v$, with one state $s_v \in \Theta_v$ per value in $D_v$. For every action $a \in \mathcal{A}$, there is an outgoing transition from $s_v$ if $v \notin \mathcal{V}(pre_a)$ or $pre_a(v) = s_v$ which leads to $s_v$ if $v \notin \mathcal{V}(e\!f\!f_a)$ or $t_v$ if $e\!f\!f_a(v) = t_v$.

As running example, consider a task where a truck can drive between four locations with a limited amount of fuel and with the restriction that the engine can only be turned on with a full tank. This can be encoded as an FDR task with three variables $\mathcal{V} = \{v_t, v_f, v_s\}$ with domains $D_t = \{A, B, C, D\}$, $D_f = \{2, 1, 0\}$, and $D_s = \{off, rd, on\}$ that represent the position of the truck, the amount of fuel available, and the status of the engine (off, ready, on), respectively. In the atomic FTS task, shown in Fig. 1a, there are hence three TSs $\Theta^{v_t}, \Theta^{v_f}, \Theta^{v_s}$, one for each variable. The task has an action $DR_{x\text{-}y, f_1\text{-}f_2}$ with precondition $\{v_t = x, v_f = f_1, v_s = on\}$ and effect $\{v_t = y, v_f = f_2\}$ for every pair of connected locations $(x, y)$, and every $f_1, f_2 \in D_f$ s.t. $f_2 = f_1 - 1$. These actions induce transitions from $x$ to $y$ in $\Theta^{v_t}$, from $f_1$ to $f_2$ in $\Theta^{v_f}$, and a self-looping transition at state on in $\Theta^{v_s}$. Furthermore, there exist actions check-fuel, CF, with precondition $\{v_f = 2, v_s = off\}$ and effect $\{v_s = rd\}$ and ON with precondition $\{v_s = rd\}$ and effect $\{v_s = on\}$. All actions have unit cost. The initial state of the FDR task is $s^{\mathcal{I}} = \{v_t = A, v_f = 2, v_s = off\}$ and its goal

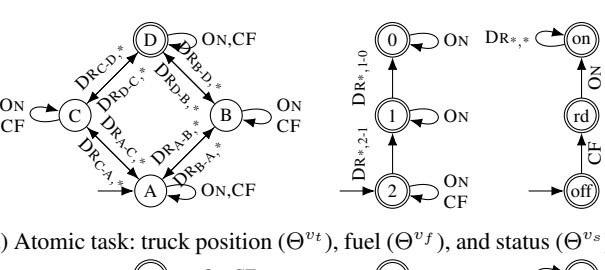

(a) Atomic task: truck position ($\Theta^{v_t}$), fuel ($\Theta^{v_f}$), and status ($\Theta^{v_s}$).

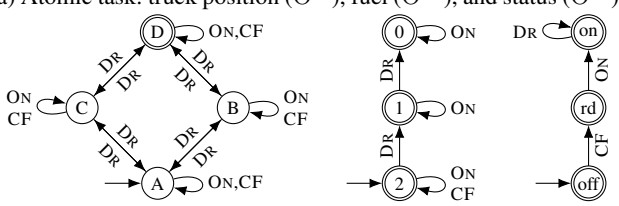

(b) After label reduction

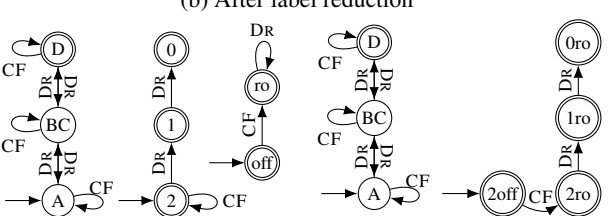

(c) After shrinking and removing irrelevant label ON.    (d) After merging and pruning unreachable states.

Figure 1: Example FTS task where a truck must drive from $A$ to $D$ with a fuel capacity of 2 and the restriction to first check the fuel capacity and turn on the engine. Transitions with wildcards (*) have multiple labels, e.g., $DR_{A\text{-}B,*}$ stands for $DR_{A\text{-}B,2\text{-}1}$ and $DR_{A\text{-}B,1\text{-}0}$. Each subfigure corresponds to a reformulation (see Section ).

is $\mathcal{G} = \{v_t = D\}$, which translates to $(A, 2, off)$ being the initial state (marked with incoming arrows) and all $(D, *, *)$ being goal states (marked with double circles) of the FTS task.

The reverse transformation from an FTS to an FDR task is not as straightforward and it may require to introduce more FDR actions than there are labels in the FTS task. The reason is that transitions in the individual TSs are more expressive than the precondition-effect tuple of FDR actions because they can encode a limited form of angelic non-determinism, disjunctive preconditions, and conditional effects. Consider the task shown in Fig. 1b, an FTS task of the same planning task that uses label $DR$ for all drive actions. Translating this task to FDR requires re-introducing multiple actions to represent $DR$ for different pairs of locations and amounts of fuel. One reason is the non-determinism where there are multiple transitions with the same label and source state, but different targets. For example, in the state $(A, 2, on)$, we can apply two transitions with label $DR$ to reach either $(C, 1, on)$ or $(B, 1, on)$. The non-determinism is angelic because the result is chosen by the planner at will. Also, the transitions in $\Theta^{v_f}$ encode a disjunctive precondition ($DR$ is applicable for $v_f = 2$ or $v_f = 1$) and conditional effects (the result of $DR$ is $v_f = 1$ iff $v_f = 2$ holds in the source state).

# M&S Task Reformulation Framework

A task reformulation is a transformation of a task such that any solution for the new task can be transformed into a solution for the original task. We follow the definition of task reduction introduced by Tozicka et al. (2016) but without requiring the reformulated task to be smaller than the input task. Most of the reformulations we consider aim to reduce the size of the task, but reformulations that make the task bigger may be useful as well, e.g., if it makes the search space smaller.

**Definition 1** (Task reformulation). *A task reformulation $\rho$ is a partial function from tasks to tasks s.t.:*

1. *$\rho(\Pi)$ is solvable if and only if $\Pi$ is solvable, and*
2. *there exists a plan reconstruction function $\overleftarrow{\rho}$ that maps each solution $\pi$ of $\rho(\Pi)$ to a solution $\overleftarrow{\rho}(\pi)$ of $\Pi$.*

A task reformulation is *polynomial* if both $\rho$ and $\overleftarrow{\rho}$ can be computed in polynomial time in the size of the input task and the reconstructed plan. It is *optimal* if, given an optimal plan $\pi$ of $\rho(\Pi)$, $\overleftarrow{\rho}(\pi)$ is an optimal plan of $\Pi$. We are interested in polynomial reformulations for optimal and satisficing planning. Note that we explicitly allow the reformulated plan to be exponentially larger than the input task. This is necessary for domains (e.g. Towers of Hanoi) where the original plan is exponentially long, but a reformulation with a solution that implicitly encodes the plan can be found in polynomial time.

## Merge-and-Shrink Transformations

There are multiple M&S transformations that can be used to reformulate an FTS task $\Pi^{\mathcal{T}} = \{\Theta_1, \ldots, \Theta_n\}$ with labels $L$. A transformation is *exact* if it preserves the set of solutions and hence is an optimal reformulation.

*Label reduction* reduces the set of labels by mapping some of them to a common new one (Sievers, Wehrle, and Helmert 2014). It is exact if for any pair of labels $\ell, \ell' \in L$ reduced to the same label, $c(\ell) = c(\ell')$ and $\ell$ and $\ell'$ induce the same transitions in all but (at most) one $\Theta_i$, $1 \leq i \leq n$. The task of Fig. 1b is the result of repeatedly applying exact label reduction on the atomic task of Fig. 1a. By itself, it does not affect the search space, but it reduces the amount of labels increasing the efficiency and effectiveness of other transformations.

*Shrinking* consists of replacing one TS $\Theta_i \in \Pi^{\mathcal{T}}$ by an abstraction thereof. This results in an abstraction of the original task, possibly introducing spurious plans that do not have any counterpart in the original task. Therefore, not all shrink transformations are suitable for task reformulation. However, using refinable abstraction hierarchies is a long standing idea in planning (Sacerdoti 1974; Bacchus and Yang 1994; Knoblock 1994). We compute refinable abstractions via shrinking strategies based on bisimulation (Milner 1971).

**Definition 2** (Bisimulation). *Let $\Theta = \langle S, L, T, s^I, \mathcal{S}^\star \rangle$ be a TS. An equivalence relation $\sim$ on $S$ is a goal-respecting bisimulation iff $s \sim t$ implies that (a) $s \in \mathcal{S}^\star \leftrightarrow t \in \mathcal{S}^\star$, and (b) $\{[s'] \mid s \xrightarrow{\ell} s' \in T\} = \{[t'] \mid t \xrightarrow{\ell} t' \in T\}$ for all $\ell \in L$ where $[s]$ denotes the equivalence class of $s$.*

Bisimulation shrinking aggregates all states in the same equivalence class of the coarsest bisimulation of some $\Theta_i \in \Pi^{\mathcal{T}}$. This is a symmetry-reduction technique that preserves all plans and as such is an exact transformation (Helmert et al. 2014; Sievers et al. 2015). In our example (cf. Fig. 1b), states B and C of $\Theta^{v_t}$ are bisimilar in $\Theta^{v_t}$ and are hence combined into a new state BC by bisimulation shrinking (cf. Fig. 1c). Note that shrinking $B$ and $C$ is only possible after label reduction, since otherwise their outgoing labels differ.

When preserving optimality is not necessary, it suffices to guarantee that any abstract plan can be refined into a real plan. Hoffmann, Kissmann, and Torralba (2014) used shrinking strategies with this property for proving unsolvability with M&S. We re-define these strategies using a different nomenclature based on the notion of weak bisimulation (Milner 1971; 1990). The key idea is to consider $\tau$-labels which are "internal" to a TS in the sense that they can always be taken in $\Theta_i$ without changing other TSs. The set of $\tau$-labels for $\Theta_i$ consists of those labels $\ell$ having a transition $s_j \xrightarrow{\ell} s_j \; \forall s_j \in \Theta_j \; \forall \Theta_j, j \neq i$. Other definitions are possible; ours is more general than that of own-labels used by Hoffmann, Kissmann, and Torralba (2014), whereas there are stronger notions based on dominance (Torralba 2017; 2018). We use $\overset{\tau}{\Longrightarrow}$ to denote a (possibly empty) path using only $\tau$-labels, and $s \overset{\ell}{\Longrightarrow} s'$ as a shorthand for $s \overset{\tau}{\Longrightarrow} \overset{\ell}{\rightarrow} \overset{\tau}{\Longrightarrow} s'$.

Following the observation by Haslum (2007) that it suffices to focus on paths with labels that either are outside relevant (i.e., have some effect on other variables) or reach the goal, we devise a variant of weak bisimulation that ignores some irrelevant paths. We say that a label $\ell$ is outside relevant for a transition system $\Theta_i$ if there exists some $\Theta_j$ with $i \neq j$ such that $s_j \xrightarrow{\ell} t_j$ for some $s_j \neq t_j$. A path $s_i \overset{\ell}{\Longrightarrow} s_i'$ is *relevant* for $\Theta_i$ if $\ell$ is outside relevant for $\Theta_i$, or there does not exist $s_i \overset{\tau}{\Longrightarrow} s_i''$ such that $s_i' \sim s_i''$. Otherwise, it is safe to ignore such path in weak bisimulation because the alternative $\tau$-path can always be used to reach $s''$ instead.

**Definition 3** (Weak Bisimulation). *Let $\Theta$ be a TS with a set $\tau$ of $\tau$-labels, and a set $T_{rel}$ of relevant paths. An equivalence relation $\sim$ on $S$ is a goal-respecting weak bisimulation iff $s \sim t$ implies $(\exists_{s' \in \mathcal{S}^\star} s \overset{\tau}{\Longrightarrow} s') \leftrightarrow (\exists_{t' \in \mathcal{S}^\star} t \overset{\tau}{\Longrightarrow} t')$, and $\forall_{\ell \in L} \{[s'] \mid s \overset{\ell}{\Longrightarrow} s' \in T_{rel}\} = \{[t'] \mid t \overset{\ell}{\Longrightarrow} t' \in T_{rel}\}$.*

Weak bisimulation shrinking maps all weakly bisimilar states into the same abstract state. In our example (cf. Fig. 1b), ON is a $\tau$-label in $\Theta^{v_s}$, therefore states rd and on of $\Theta^{v_s}$ are weakly bisimilar (both have a single relevant path $\overset{\text{DR}}{\Longrightarrow} [\text{on}]$) resulting in $\Theta^{v_s}$ as shown in Fig. 1c.

Another useful abstraction transformation consists of removing TSs with a *core* state. We say that a state $s^C$ is a *core* for $\Theta_i$ if (1) for every outside relevant label $\ell$ there exists $s^C \overset{\ell}{\Longrightarrow} s^C$, (2) there is a $\tau$-path from the initial state to $s^C$, and (3) there is a $\tau$-path from $s^C$ to a goal state. Such a TS can be abstracted away because all outside relevant labels can always be reached via a $\tau$-path through $s^C$.

*Merging* replaces two TSs by their synchronized product. Fig. 1d shows the FTS task that results from merging $\Theta^{v_f}$

and $\Theta^{v_s}$ of the FTS task shown in Fig. 1c. Merging is an exact transformation (Helmert et al. 2014), which comes at the price that the size of the task grows quadratically with every merge, so it increases exponentially with the number of merges. In practice, we limit the maximum size of any TS in the reformulated task, forbidding any merge that goes beyond this limit. As label reduction, by itself merging does not change the reachable search space. However, it often enables additional label reduction, shrinking, and/or pruning. In our example of Fig. 1d, CF has become a $\tau$-label, so 2off and 2ro could be reduced by weak bisimulation.

Finally, there are multiple *pruning* techniques defined in the M&S framework. If a state $s_i$ is *unreachable* (from the initial state) or *irrelevant* (cannot reach a goal) in any $\Theta_i$, it can be pruned (Helmert et al. 2014). If a label $\ell$ is *dead* (i.e., there is no transition labeled with $\ell$ in any $\Theta_i \in \Pi^{\mathcal{T}}$) or *irrelevant* (i.e., all transitions labeled with $\ell$ are self-loop transitions), then it can be pruned (Sievers, Wehrle, and Helmert 2014). If a TS $\Theta_i \in \Pi^{\mathcal{T}}$ is the only one with a goal defined, i.e., there are no non-goal states in $\Theta_j \in \Pi^{\mathcal{T}}$ with $j \neq i$, all outgoing transitions from goal states in $\Theta_i$ can be removed (Hoffmann, Kissmann, and Torralba 2014). If a TS has only one state and no dead labels, it can be pruned. All these pruning techniques preserve at least one optimal plan and are therefore exact transformations.

## Plan Reconstruction

M&S iteratively applies the transformations described above on a task $\Pi^{\mathcal{T}} = \{\Theta_1, \ldots, \Theta_k\}$, resulting in a sequence of reformulation steps $\rho_1, \ldots, \rho_n$ producing a sequence of planning tasks $\Pi_0^{\mathcal{T}}, \ldots, \Pi_n^{\mathcal{T}}$ where $\Pi_0^{\mathcal{T}} = \Pi^{\mathcal{T}}$, and $\Pi_i^{\mathcal{T}} = \rho_i(\Pi_{i-1}^{\mathcal{T}})$ for $i \in [1, n]$. We can run any planning algorithm to find a plan $\pi^{\rho_n} = s_1^{\rho_n} \xrightarrow{\ell_1^{\rho_n}} s_2^{\rho_n} \xrightarrow{\ell_2^{\rho_n}} s_3, \ldots$ of the final task $\Pi_n^{\mathcal{T}}$. The plan reconstruction procedure is then tasked to compute a plan $\pi = s_1 \xrightarrow{\ell_1} s_2 \xrightarrow{\ell_2} \ldots$ for the original task $\Pi^{\mathcal{T}}$ from $\pi^{\rho_n}$ and the sequence of reformulations.

Performing a reconstruction $\overleftarrow{\rho_i}$ for each step $\rho_i$ has some overhead because it requires to store each intermediate task. We avoid this by combining sequences of reformulations that correspond to merge, label reduction, and bisimulation transformations. Pruning-based transformations can be ignored by the plan reconstruction procedure because the plan found is still valid for the original task without any modifications. Plan reconstruction can be done for the entire transformation at once without storing information about the intermediate planning tasks. Therefore, we have a sequence of transformations $\Pi^{\mathcal{T}} \xrightarrow{\rho_1} \Pi_1^{\mathcal{T}} \xrightarrow{\rho_2} \Pi_2^{\mathcal{T}} \ldots$ with only two types of reformulations to consider: merging + label reduction + bisimulation shrinking ($\rho^{MLB}$), and weak bisimulation shrinking ($\rho^{\tau B}$).

We first consider the reconstruction of a reformulation $\rho^{MLB}$ on a task $\Pi_i^{\mathcal{T}}$, resulting in a task $\Pi_{i+1}^{\mathcal{T}}$. The state space of $\Pi_{i+1}^{\mathcal{T}}$ is a bisimulation of the state space of $\Pi_i^{\mathcal{T}}$, so any sequence $s_1 \xrightarrow{\ell_1} s_2 \xrightarrow{\ell_2} \ldots$ in $\Pi_i^{\mathcal{T}}$ has its counterpart $\alpha(s) \xrightarrow{\ell_1'} \alpha(s_2) \xrightarrow{\ell_2'} \ldots$ in $\Pi_{i+1}^{\mathcal{T}}$ and vice versa. To reconstruct the plan, we need two functions $\alpha$ and $\lambda$, mapping states and labels in $\Pi_i^{\mathcal{T}}$ to states and labels in $\Pi_{i+1}^{\mathcal{T}}$. The $\alpha$ function is computed by M&S heuristics and

compactly represented with the so-called cascading tables or merge-and-shrink representation (Helmert et al. 2014; Helmert, Röger, and Sievers 2015). The label mapping is simply the composition of all label reduction transformations used by $\rho^{MLB}$.

The plan can be reconstructed step by step, starting from $s^{\mathcal{I}}$. Given the current factored state $s$ and a step in the abstract plan $\alpha(s) \xrightarrow{\ell'} t'$, find a transition $s \xrightarrow{\ell} t$ such that $\alpha(t) = t'$ and $\lambda(\ell) = \ell'$. Note that the straightforward approach of enumerating all transitions applicable from $s$ is not guaranteed to terminate in polynomial time because, unlike in FDR tasks where the number of successors is bounded by the number of actions, in FTS there may be exponentially many successors in the size of the task. However, one can use the cascading tables representation to retrieve a factored state $t = (t_1, \ldots, t_n)$ such that $s \xrightarrow{\ell} t$, $\ell' = \lambda(\ell)$ and $\alpha(t) = t'$. This works as follows: First, for each transition system $\Theta_i$, obtain the set $S_i'$ of target states $t_i$ such that $s_i \xrightarrow{\ell} t_i$ for any label $\ell$ such that $\ell' = \lambda(\ell)$. Then, traverse the cascading tables and, for each intermediate table that maps states of two transition systems $\Theta_i, \Theta_j$ to an abstract TS $\Theta_\gamma = \gamma(\Theta_i \otimes \Theta_j)$, compute the set of abstract states $S_\gamma = \{s_\gamma \mid \exists_{s_i \in S_i', s_j \in S_j'} \ s_\gamma = \gamma((s_i, s_j))\}$, mapping each $s_\gamma \in S_\gamma$ to one such $(s_i, s_j)$ pair. This allows us to keep track of one factored state for each abstract state. After all cascading tables have been traversed, it suffices to return the factored state $t$ associated with the abstract state $t'$.

**Proposition 1.** *Label reduction, merging (up to a size limit), pruning and bisimulation shrinking are optimal and polynomial reformulations.*

*Proof.* It is well-known that all these techniques can be computed in polynomial time (Helmert et al. 2014; Sievers, Wehrle, and Helmert 2014). Each step of the plan can be reconstructed by traversing the cascading-tables representation, which is polynomial in the size of the input task. $\square$

We now consider the reconstruction of a reformulation $\rho^{\tau B}$ on a task $\Pi_i^{\mathcal{T}} = \{\Theta_1, \ldots, \Theta_k\}$ where $\rho^{\tau B}$ applies weak bisimulation shrinking to some TS in $\Pi_i^{\mathcal{T}}$. We assume WLOG that $\Theta_1$ is the shrunk TS, so $\Pi_{i+1}^{\mathcal{T}} = \{\alpha^{\tau B}(\Theta_1), \Theta_2, \ldots, \Theta_k\}$. As $\alpha^{\tau B}$ is induced by a weak bisimulation on the states of $\Theta_1$, then for any state $s$ in $\Pi_i^{\mathcal{T}}$ and any transition $\rho^{\tau B}(s) \xrightarrow{\ell} t^\rho$ in the reformulated task, there exists a path $s \xRightarrow{\ell} t$ in the original task such that $\rho^{\tau B}(t) = t^\rho$. Therefore, to reconstruct the plan for $\Pi_i^{\mathcal{T}}$ from a plan for $\Pi_{i+1}^{\mathcal{T}}$ one must re-introduce the $\tau$-label transitions until reaching a state where $\ell$ is applicable and this results in some $t$ such that $\rho^{\tau B}(t) \xRightarrow{\tau} t^\rho$. The search can be done locally in $\Theta_1$ because $\tau$-labels have self-loop transitions in other TSs. To do so, we first look for all states $u_1$ such that $u_1 \xrightarrow{\ell} (\xrightarrow{\tau})^* t_1$ in $\Theta_1$ and $(\alpha^{\tau B}(t_1), s[\Theta_2], \ldots, s[\Theta_n]) = t^\rho$. Then, we run uniform-cost search from $s[\Theta_1]$ using only transitions with $\tau$-labels until we reach such an $u_1$. Note that this runs in polynomial time in the size of the input task.

This procedure has similarities with red-black plan repair (Domshlak, Hoffmann, and Katz 2015), the plan reconstruction of the merge values reformulation (Tozicka et

al. 2016), or decoupled search (Gnad and Hoffmann 2018). These algorithms repair an abstract/relaxed plan by introducing additional actions to enable the preconditions ignored by the relaxed plan. Our case is slightly more complex because the same label may have multiple targets so one must ensure the remaining abstract plan is applicable in the resulting state.

If a TS $\Theta_i$ with a core state $s^C$ was abstracted away, its corresponding path must be reconstructed as well. For each transition in the abstract plan with label $\ell$, we find the shortest $s_i \overset{\ell}{\Longrightarrow} s_i'$ from the current state $s_i$ (initialized to the initial state of $\Theta_i$ in the first iteration, and to the final state in the path of the previous iterations afterwards), and $s_i' \overset{\tau}{\Longrightarrow} s^C$. Note that to keep the plan shorter, we do not enforce the $\tau$-path to go via the core state, but rather the condition above suffices to ensure that the rest of the plan can be reconstructed.

**Proposition 2.** *Weak bisimulation shrinking is a polynomial reformulation.*

*Proof.* The coarsest weak bisimulation of a TS can be computed by computing the bisimulation of the transitive closure of the TS over $\tau$. Each step of the plan reconstruction corresponds to an uniform-cost search on each TS. Both operations take polynomial time in the size of the TS. □

Consider the following plan of the task shown in Fig. 1c: $(A, 2, \text{off}) \overset{\text{CF}}{\longrightarrow} (A, 2, \text{ro}) \overset{\text{DR}}{\longrightarrow} (BC, 1, \text{ro}) \overset{\text{DR}}{\longrightarrow} (D, 0, \text{ro})$. To reconstruct the plan for the task prior to weak bisimulation shrinking (cf. Fig. 1b), we execute it and, when DR cannot be applied in rd, we insert a $\tau$-transition with ON resulting in the plan: $(A, 2, \text{off}) \overset{\text{CF}}{\longrightarrow} (A, 2, \text{rd}) \overset{\text{ON}}{\longrightarrow} (A, 2, \text{on}) \overset{\text{DR}}{\longrightarrow} (BC, 1, \text{on}) \overset{\text{DR}}{\longrightarrow} (D, 0, \text{on})$. Then, we reconstruct the plan for the atomic task of Fig. 1a step by step, resulting in a plan: $(A, 2, \text{off}) \overset{\text{CF}}{\longrightarrow} (A, 2, \text{rd}) \overset{\text{ON}}{\longrightarrow} (A, 2, \text{on}) \overset{\text{DR}_{\text{A-B,2-1}}}{\longrightarrow} (B, 1, \text{on}) \overset{\text{DR}_{\text{B-D,1-0}}}{\longrightarrow} (D, 0, \text{on})$.

## Relation to FDR Reformulation Methods

The M&S reformulations are closely related to previous FDR reformulation methods like the generalize actions (Tozicka et al. 2016), fluent merging (Seipp and Helmert 2011), and abstraction-based reformulations (Helmert 2006b; Haslum 2007; Tozicka et al. 2016). To compare reformulation methods over different formalisms, we consider that a method dominates another if it can perform the same reformulations.

**Definition 4** (Dominance of Reformulation Methods). *An FTS task reformulation method $X$ dominates an FDR reformulation method $Y$ if, given an FDR task $\Pi^{\mathcal{V}}$ and a reformulation $\rho^Y \in Y$ applicable over $\Pi^{\mathcal{V}}$, there exists a reformulation $\rho^X \in X$ such that it is applicable in $atomic(\Pi^{\mathcal{V}})$ and $\rho^X(atomic(\Pi^{\mathcal{V}})) = atomic(\rho^Y(\Pi^{\mathcal{V}}))$. We say that the domination is strict if there exists $\rho^X \in X$ such that it is applicable in $atomic(\Pi^{\mathcal{V}})$ but there does not exist any $\rho^Y \in Y$ applicable in $\Pi^{\mathcal{V}}$ and $\rho^X(atomic(\Pi^{\mathcal{V}})) = atomic(\rho^Y(\Pi^{\mathcal{V}}))$.*

The *generalize actions* reformulation reduces the number of FDR actions by substituting two actions by a single one if they are equal except for a precondition on a binary variable. Formally, whenever there is a variable $w$ with domain $D_w = \{x, y\}$, and two actions $a_1, a_2$ s.t. $\mathcal{V}(pre_{a1}) = \mathcal{V}(pre_{a2})$, $\forall v \in (\mathcal{V}(pre_{a1}) \setminus \{w\})$ $pre_{a1}(v) = pre_{a2}(v)$, $pre_{a1}(w) = x$, $pre_{a2}(w) = y$, and $eff_{a1} = eff_{a2}$. Then, $a_1$ and $a_2$ can be replaced by $a'$ where $eff_{a'} = eff_{a1}$ and $pre_{a'}(v) = pre_{a1}(v) \ \forall v \in (\mathcal{V}(pre_{a1}) \setminus \{w\})$.

**Theorem 1.** *Exact label reduction strictly dominates the generalize actions reformulation.*

*Proof Sketch.* If generalize actions replaces $a_1$ and $a_2$ in $\Pi^{\mathcal{V}}$ by $a'$, then there are labels $\ell_1$ and $\ell_2$ in $atomic(\Pi^{\mathcal{V}})$ that correspond to $a_1$ and $a_2$ and a TS $\Theta_w$ that corresponds to $w$ in $\Pi^{\mathcal{V}}$. As $a_1$ and $a_2$ have the same effects and preconditions on all variables except $v$, then $\ell_1$ and $\ell_2$ are equal except for $\Theta_w$ so they can be reduced. Label reduction is more general because it may result in transitions with different targets from the same state and label, which is not possible in FDR. □

*Fluent merging* is an FDR reformulation inspired by the merge transformation in M&S (Seipp and Helmert 2011). It replaces two variables $v_1, v_2 \in \mathcal{V}$ by their product, resulting in a variable $v_{1,2}$ with domain $D_{v_1, v_2} = D_{v_1} \times D_{v_2}$. However, adapting the FDR actions is not straightforward since they would require disjunctive preconditions. For example, if action $a_1$ has a precondition on $v_1$ but not on $v_2$, then the action is applicable for several values of $D_{v_1, v_2}$ but not for all of them. Since FDR does not allow for disjunctive preconditions, multiple copies of the actions are needed to encode the preconditions and effects on the new variable. Similarly, auxiliary actions must be added to encode a disjunctive goal whenever a goal and a non-goal variable are merged. In this case, the merge transformation does not dominate fluent merging because it does not add such auxiliary labels and transitions. This is arguably an advantage since adding them is not expected to be beneficial or, otherwise, an equivalent reformulation could be defined in M&S.

The use of abstraction for task reformulation in planning has a long history (Knoblock 1994). The key idea is to solve an abstraction of the problem and then refine the abstract solution by filling the gaps. Not all abstractions are suitable for this, since they need to ensure that any solution for the abstract task can be refined into a plan for the original task. Abstractions with this property are said to be *refinable*. Abstraction reformulations were first applied in FDR by the Fast Downward planner (Helmert 2006a). Their reformulation abstracts away any root variable in the causal graph (i.e., does not have dependencies on other variables) whose free domain transition graph is strongly connected (i.e., one can always set the variable to any desired value by applying a sequence of actions). This was generalized by Haslum (2007) into the *safe variable abstraction* reformulation under the observation that (1) it suffices to consider values that are relevant for other variables (because they are precondition or effect of an action that has another variable in its effect); and (2) the goal only needs to be achieved at the end of the plan so the goal value must be free reachable from other relevant values, but it is not necessary that other values are reachable from the goal value.

Finally, one can also ignore the difference among some values of a variable without ignoring it completely: the *merge values* reformulation reduces the domain of an FDR variable by merging several values whenever they can be switched via actions without any side effects (Tozicka et al. 2016). Formally, let $v$ be a variable with $x, y \in D_v$, and $a_1$ and $a_2$ be actions s.t. $\mathcal{V}(pre_{a1}) = \mathcal{V}(\textit{eff}_{a1}) = \mathcal{V}(pre_{a2}) = \mathcal{V}(\textit{eff}_{a2}) = \{v\}$, and $pre_{a1}(v) = \textit{eff}_{a2}(v) = x$, and $pre_{a2}(v) = \textit{eff}_{a1}(v) = y$. Then, $x$ may be removed from $D_v$, replacing every occurrence of $x$ in $A$, $I$, and $G$ by $y$.

As all these methods, weak bisimulation shrinking obtains a refinable abstraction, but on the FTS representation, taking advantage of the flexibility of M&S to compute abstractions.

**Theorem 2.** *Removing transition systems with core states after applying weak bismulation shrinking strictly dominates the safe variable abstraction reformulation.*

*Proof Sketch.* If a variable is abstracted away, abstract states corresponding to the values that appear in the preconditions of outside relevant actions are all weakly bisimilar. After shrinking, the resulting abstract state is a core state. □

**Theorem 3.** *Weak bisimulation shrinking strictly dominates the merge values reformulation.*

*Proof Sketch.* If values $x, y$ of variable $v$ are merged, there exist $\ell_1, \ell_2$ in $atomic(\Pi^{\mathcal{V}})$ corresponding to $a_1, a_2$, and a TS $\Theta_v$ representing $v$. As $v$ is the only variable in the preconditions and effects of $a_1$ and $a_2$, $\ell_1$ and $\ell_2$ are $\tau$-labels in $\Theta_v$. Since $x \xrightarrow{\tau} y$ and $y \xrightarrow{\tau} x$, $x$ and $y$ are weakly bisimilar. □

## Search on the FTS Representation

To use our reformulation framework, planning algorithms must be used to find a solution to the reformulated FTS task. Heuristic search is a leading approach for solving classical planning problems (Bonet and Geffner 2001). A compilation into an FDR task having an action for each combination of transitions with the same label in different TSs is possible, but may incur a big overhead, potentially losing any gains obtained by the reformulation methods. Here, we consider how to apply heuristic search algorithms to FTS tasks by defining the successor generation and heuristic evaluation.

Successor generation is the operation that, given a state $s$, generates all transitions $s \xrightarrow{l} t$ in the state space of the task. This typically is done in two steps: (1) generate the set of actions that are applicable in $s$ and (2) for each such action obtain the corresponding successor state.

Since the number of actions in FDR tasks may be very large, iterating over all of them to check whether they are applicable in $s$ is inefficient. The Fast Downward Planning System uses a tree data-structure to efficiently retrieve the applicable actions in a given state (Helmert 2006b). However, this data-structure relies on actions being applicable either only for one value of each variable if $v \in \mathcal{V}(pre_a)$ or in all values of such variable otherwise. This is no longer true for labels in the FTS representation. A label is applicable in a factored state $s$ if there exists an outgoing transition $s[\Theta_i] \xrightarrow{l} t_i$ for each $\Theta_i \in \Pi^{\mathcal{T}}$. Since there may be

any number of transitions in each $\Theta_i$ from any number of source states, labels may be applicable for arbitrary sets of states. We pre-compute for every abstract state $s_i \in \Theta_i$ the set of labels with an outgoing transition from $s_i$, denoted $L_{s_i}$. Then, given a state $s$, the set of applicable labels can be computed as $\bigcap_{\Theta_i} L_{s[\Theta_i]}$.

Step (2) is simple in FDR since the new state is a copy of $s$, overriding the value of variables in the effect. In the FTS representation, however, there may be multiple successors from $s$ with label $\ell$. We enumerate all possible successors by considering all outgoing transitions from $s[\Theta_i]$ in every $\Theta_i$. To do this efficiently, for each label $\ell$ we divide the set of TSs in $\Pi^{\mathcal{T}}$ in three sets: the irrelevant TSs where $\ell$ only induces self-loop transitions, deterministic TSs where for every $s_i \in \Theta_i$ there is a single outgoing transition with $\ell$, and non-deterministic TSs where there may be multiple transitions from the same source state. Only the latter require to enumerate all possible transitions, whereas irrelevant TSs are ignored and the effect on deterministic TSs can be set as in FDR tasks.

We now discuss how to derive heuristic functions for the FTS representation, which are essential to guide the search and find solutions to large tasks. As most heuristic functions have originally been defined for STRIPS or FDR, they need to be adapted to use them in FTS tasks. This is similar to adding support for a limited form of disjunctive preconditions and conditional effects. In optimal planning, we use merge-and-shrink heuristics since they are already based on FTS.

To apply our reformulation framework on satisficing planning, we adapt the FF heuristic (Hoffmann and Nebel 2001). FF is based on the delete-relaxation, ignoring the delete effects of STRIPS actions. In FDR, "ignoring deletes" is interpreted as ignoring the negative effect of the actions, so that variables accumulate values instead of replacing them. This is easily extrapolated to the FTS representation by considering that each TS may simultaneously be in multiple states.

To compute the heuristic, we compile our task into an FDR task with one unary action $a_{s_i, \ell, t_i}$ for each transition $s_i \xrightarrow{\ell} t_i$ in some $\Theta_i$. This action has $t_i$ as effect, and $s_i$ as precondition plus additional preconditions for each other $\Theta_j$ where $l$ is not applicable in all states. If there is a single state $s_j \in \Theta_j$ where $\ell$ is applicable, we add $s_j$ to the precondition of $a_{s_i, \ell, t_i}$. If there are more than one, we add an auxiliary fact to our task $f_{\ell, j}$ that represents the disjunction of those states, as well as auxiliary unary actions from each of those states to $f_{\ell, j}$.

Afterwards, we retrieve the relaxed plan as a set of transitions $s_i \xrightarrow{\ell} t_i$, and add the cost of all their labels to obtain the heuristic value. One difference to FF for FDR is that there, FF counts each action only once because no action needs to be applied more than once in delete-free tasks. We do not do this to avoid underestimating the goal distance when the same label may have different effects (e.g. label DR in Fig.1b).

The delete-relaxation is also useful to select preferred actions. In FDR, an action is preferred in state $s$ if it belongs to the relaxed plan of FF for $s$. In FTS, we consider $s \xrightarrow{\ell} t$ to be preferred if the relaxed plan from $s$ contains a transition

labeled with $\ell$ and with target $t_i$ s.t. $\exists \Theta_i t[\Theta_i] = t_i$.

# Experiments

We implemented the M&S reformulation framework in Fast Downward (FD) (Helmert 2006b), using its existing M&S framework (Sievers 2018) and extending it with weak bisimulation as well as pruning transformations that remove dead labels and irrelevant TSs and labels. We also modified the layout of the algorithm: firstly, since our pruning transformations might trigger further pruning opportunities, we always repeatedly apply them until a fixpoint is reached. Secondly, we run label reduction and shrinking on the atomic FTS task until no more simplifications are possible. Finally, we cannot exactly control the amount of shrinking done because this would result in non-refinable abstractions that do not admit plan reconstruction. Instead, we restrict merging to satisfy the size limit and only shrink after merging and pruning.

To consider the effects of some of the M&S transformations on the task reformulation individually, we consider the following configurations. As the simplest baseline, we only transform the FDR task (FDR) into the atomic FTS task (a), without any further transformations. This does not affect the state space at all, but serves for quantifying the overhead of our implementation over FD, mainly due to using different data structures to represent the task and perform successor generation. Another variant of atomic adds exact label reduction and shrinking (a-ls), either based on bisimulation for optimal planning or weak bisimulation for satisficing planning. Other configurations combine label reduction and shrinking with a merge strategy. For the latter, we consider DFP (d-ls) and sbMIASM (m-ls, called dyn-MIASM originally) (Sievers, Wehrle, and Helmert 2014), with a size limit of 1000 on the resulting product. We did not find qualitative differences with size limits of 100 and 10000. We impose a time limit of $900s$ on the reformulation process. For the overall planning, we use a limit of 3.5 GiB and $1800s$. We use all STRIPS benchmarks from the optimal/satisficing tracks of all IPCs, two sets consisting of 1827/1816 tasks across 48 unique domains.[1]

## Search Space Reduction

To assess the impact of our task reformulations on the reachable state space, we run uniform-cost search and evaluate the number of expansions until the last $f$-layer. Fig. 2 compares the FDR representation against a-ls and d-ls with bisimulation (top) and weak bisimulation (bottom) shrinking. We observe that even with only label reduction and bisimulation shrinking (a-ls) there are state space reductions of up to one order of magnitude in some cases. Most of these gains are due to shrinking, given that label reduction does not change the state space and pruning cannot be performed often in the atomic representation due to the preprocessing of FD. When using merge transformations (d-ls), state space reductions can often be of up to several orders of magnitude. It is worth

    [1]Implementation: https://doi.org/10.5281/zenodo.3232878, dataset with benchmarks: https://doi.org/10.5281/zenodo.3232844.

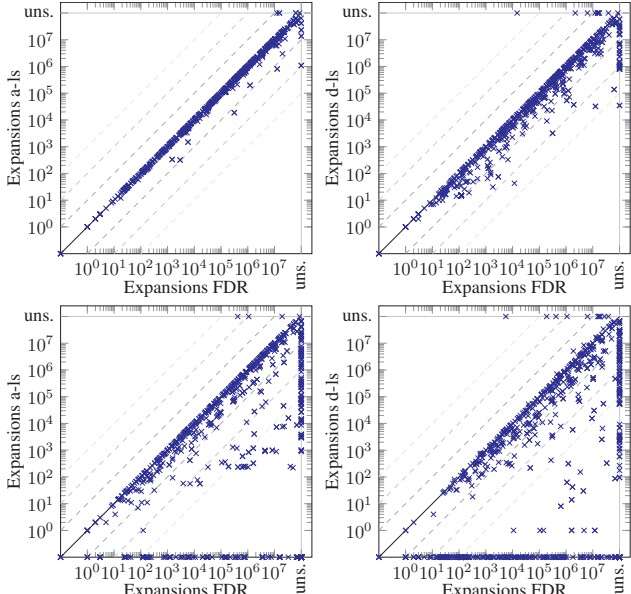

Figure 2: Expansions until last $f$-layer of blind search on the FDR task and different reformulated FTS tasks, using bisimulation (top) and weak bisimulation (bottom) for shrinking.

noting that merging does not affect the state space, so this reduction is due to the synergy with pruning and shrinking.

If optimality does not need to be preserved, larger reductions can be achieved with weak bisimulation shrinking. In this case, 305 tasks (including entire domains like logistics, miconic, movie, rovers, and zenotravel) can be solved during the reformulation resulting in 0 expansions (points on the x-axis). The reason is that weak bisimulation shrinks away entire TSs (e.g., if they form a single connected component with actions without side preconditions or effects, which translate to $\tau$-labels). An example is logistics: as trucks/airplanes can always freely change their location with the drive/fly action, weak bisimulation simplifies the TSs describing their position, after which the TSs for packages can also be simplified. Previous abstraction reformulation approaches solved many of these domains too, with the exception of Rovers, where they obtained reductions but without completely simplifying the domain. With merge reformulations, 460 tasks are solved with DFP (completely solving transport-opt), and 514 with MIASM (solving all but two instances in parcprinter-opt). This is remarkable given the low limit of 1000 abstract states.

## Results with Informed Search

We evaluate the impact of our reformulations in terms of coverage (see Table 1), expansions, and total time (see Fig. 3). On the optimal benchmarks, we run A* with $h^{\max}$ and M&S with DFP using a 50000 size limit and (approximate) bisimulation shrinking. On the satisficing benchmarks, we run lazy greedy search with $h^{\mathrm{FF}}$, with and without preferred operators. The comparison of FDR and atomic (a) shows that our implementation has some overhead. Both

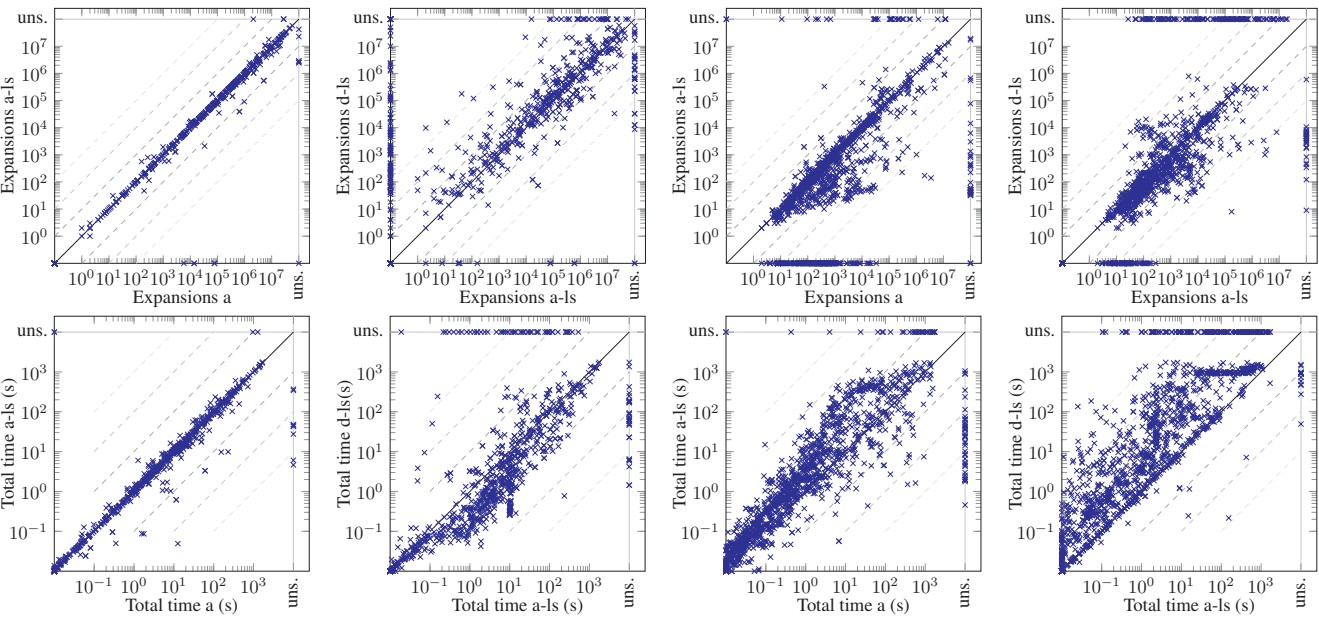

Figure 3: Expansions until last $f$-layer and total time of a vs. a-ls (left) and a-ls vs. d-ls (right) for A$^*$ with M&S (left block) and lazy greedy search with $h^{\mathrm{FF}}$ and preferred operators (right block).

configurations explore the same state space with very similar heuristics. $h^{\mathrm{max}}$ and $h^{\mathrm{FF}}$ are computed in the same way with no big overhead and the runtime of plan reconstruction is usually negligible. In terms of heuristic value, $h^{\mathrm{max}}$ is identical and $h^{\mathrm{FF}}$ only differs due to tie-breaking and because some actions may be counted twice. One of the main sources of overhead is the memory used to represent FTS tasks. Our data structures use $O(|L|)$ memory on each TS, whereas in FDR no memory is wasted for variables not mentioned in the preconditions or effects.

Label reduction and shrinking on the atomic FTS task (a-ls) is useful in most cases, increasing total coverage in all configurations. This reformulation reduces the state space as well as the task description size (i.e. reducing the TSs in the FTS representation). Therefore, gains in expanded nodes usually translate into lower search times, and it can pay off despite the overhead of the precomputation phase on total time.

Merge reductions (d-ls), however, are oftentimes harmful in combination with delete-relaxation heuristics ($h^{\mathrm{max}}$ and $h^{\mathrm{FF}}$), due to the overhead caused by increasing the task size. Nevertheless, they can be very useful in some domains, whenever there is enough synergy with pruning (e.g. woodworking, tpp) or shrinking (e.g. childsnack). Indeed, for all heuristics we tried, merge reformulations are useful in at least a few domains. This is also reflected in the orcl column that shows how many instances are solved by any of our configurations. This is often much larger than our atomic configuration, but also than the FDR baseline, showing that if the right reformulations are chosen for each domain, they can compensate for the overhead of using an FTS representation.

The results of d-ls with M&S heuristics are different be-

**A$^*$ with $h^{\mathrm{max}}$ (top) and M&S (bottom):**

| | FDR | a | a-ls | d-ls | m-ls | tot | orcl |
|---|---|---|---|---|---|---|---|
| FDR | – | **12** | **13** | **37** | **36** | 797 | |
| a | 1 | – | 1 | **36** | **36** | 770 | $h^{\mathrm{max}}$: 801 |
| a-ls | 3 | **4** | – | **36** | 35 | 780 | |
| d-ls | 2 | 2 | 1 | – | 7 | 600 | |
| m-ls | 4 | 4 | 4 | **19** | – | 632 | |
| FDR | – | 2 | 3 | 12 | 14 | 822 | |
| a | **4** | – | 1 | **13** | **16** | 826 | $h^{\mathrm{M\&S}}$: 910 |
| a-ls | **7** | **4** | – | **13** | **16** | 831 | |
| d-ls | **13** | 11 | 10 | – | 11 | 815 | |
| m-ls | **16** | 15 | 15 | **16** | – | 849 | |

**Lazy greedy search with $h^{\mathrm{FF}}$, without (top) and with (bottom) preferred operators:**

| | FDR | a | a-ls | d-ls | m-ls | tot | orcl |
|---|---|---|---|---|---|---|---|
| FDR | – | **18** | 15 | **27** | **22** | 1326 | |
| a | 6 | – | 13 | **28** | **22** | 1272 | $h^{\mathrm{FF}}$: 1413 |
| a-ls | **18** | **15** | – | **31** | **24** | 1368 | |
| d-ls | 10 | 10 | 4 | – | 11 | 1208 | |
| m-ls | 13 | 15 | 7 | **21** | – | 1224 | |
| FDR | – | **17** | **15** | **24** | **23** | 1502 | |
| a | 8 | – | **11** | **25** | **24** | 1461 | $h^{\mathrm{FF}}$ p.: 1589 |
| a-ls | 13 | 8 | – | **26** | **26** | 1471 | |
| d-ls | 9 | 6 | 2 | – | 15 | 1357 | |
| m-ls | 9 | 7 | 3 | **16** | – | 1322 | |

Table 1: Domain comparison of coverage for A$^*$ (left) with $h^{\mathrm{max}}$ (top) and M&S (bottom), and lazy greedy search (right) with $h^{\mathrm{FF}}$, without (top) and with (bottom) preferred operators. A value in row x and column y denotes the number of domains where x is better than y. It is bold if this is higher than the value in y/x. Column "tot" shows total coverage and "orcl" shows the oracle, i.e., per-task maximized, coverage over our algorithms (thus excluding FDR).

cause there are more cases where the heuristic is less informed after the d-ls reformulation, increasing the number of expansions. There is also a large number of instances where the heuristic value for the initial state is perfect for a-ls whereas a large amount of search is needed with d-ls. The reason is that the options available for the merge strategy during the reformulation are reduced by the limit on abstract states, leading to different merge decisions, and possibly degrading the quality of the heuristic. However, with

M&S heuristics there is no overhead in runtime so d-ls pays off more often.

The rightmost two columns of Fig. 3 show results with $h^{\text{FF}}$ and preferred operators for satisficing planning. The reductions obtained by weak bisimulation shrinking are much stronger than by optimality preserving strategies, improving the performance of a-ls and d-ls in terms of expanded nodes. In terms of runtime, a-ls is useful in many cases despite the overhead caused by spending up to 900s in preprocessing. Merge reformulations, however, increase the computational cost of the heuristic, so they do not pay off over a-ls except in a few cases where the reduction is huge.

## Conclusion

In this work, we use the M&S framework for task reformulation and analyze its advantages over reformulations in FDR. Our results show a large potential of state space reductions, that sometimes can solve entire domains without any search.

The framework has even more potential by integrating new reformulation methods like subsumed transition pruning (Torralba and Kissmann 2015), or graph factorization (Wehrle, Sievers, and Helmert 2016). Our results also show that not all reformulations are always helpful. Thus, to materialize all this potential, methods to automatically select the best reformulation method for each domain are also of great interest (Gerevini, Saetti, and Vallati 2009; Fuentetaja et al. 2018).

## Acknowledgments

Silvan Sievers has received funding for this work from the European Research Council (ERC) under the European Union's Horizon 2020 research and innovation programme (grant agreement no. 817639). Álvaro Torralba has received support by the German Research Foundation (DFG), under grant HO 2169/5-1 and DFG grant 389792660 as part of TRR 248 (see `https://perspicuous-computing.science`).

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
