# OpenReview forum: "Merge-and-Shrink Task Reformulation for Classical Planning"
_icaps-conference.org/ICAPS/2019/Workshop/HSDIP_

### Official Review · AnonReviewer2 · 2019-03-29
**dense read - moderate novelty - weak accept**

**Rating:** 6
**Confidence:** 4

**Review:**

This paper revisits the idea of applying a suite of automatic simplification techniques to a planning problem in advance of solving it, with the difference that simplifications are applied to the internal factored transition system representation instead of the original modelling formalism, such as STRIPS or FDR. This is claimed to have some advantage in that the representation allows for conditional effects and certain forms of disjunctions, which allows the result of certain additional simplifications to be expressed. On the downside, the authors need to recreate the planning machinery to operate on this representation, which at current is less efficient than the well-developed planner implementation in fast downward that operates on the FDR representation.

In Section "Reconstruction of MLB Reformulations", on page 4: At the beginning of the section, the label-abstracting function is falled f^L, but later in the section it seems to change name to \lambda. Is this a typo, or is \lambda a different function? If so, what is it?

Binary variables example later in the same section: The authors claim that "...applying bisimulation shrinking, the resulting TS has still only 2 states: one where both counters are set to 1 and another where at least one counter must still be set." I don't see how this can be a bisimulation. It would mean that (0,0) ~ (0,1) ~ (1,0), but applying the action that sets the first variable in (0,0) leads to (1,0) while applying it in state (0,1) leads to (1,1), which with bisimilarity should imply that (1,0) ~ (1,1), which contradicts the first condition of bisimilarity since (1,1) is a goal state and (1,0) is not.

Perhaps the authors meant applying *weak* bisimilarity? But in that case, which are the internal labels?

In Figure 2, what is the difference between the plots in the top row and the plots in the bottom row? The labels or caption does not say anything.

In the section on search space reduction, the authors mention a number of domains that are completely solved by simplification. This was also observed in the original paper. Does the generalization of the simplifications proposed here result in any new domain being fully solved, that was not with the previous versions of the same techniques, or any individual task within a not fully solved domain?

---

### Official Review · AnonReviewer1 · 2019-04-06
**This paper was a pleasure to read**

**Rating:** 10
**Confidence:** 4

**Review:**

Brief summary of the paper:
This work discusses the transformation of planning tasks based on a finite domain representation (FDR) into tasks based on factored state transition systems (FTS). The transformation is based on the Merge&Shrink framework, which consists of various abstraction-based techniques commonly used to compute abstraction heuristics. The key idea of this work is to transform the original task via a sequence of task transformations and to provide a reconstruction procedure which allows to use a plan from the transformed task to compute a plan for the original task. The paper discusses how to perform search on these FTS-based tasks and also provides an adaptation of the FF heuristic. Additionally,  previous FDR task reformulations are put into context and it is formally shown that these are dominated by the FTS reformulations presented here. The paper concludes with a comprehensive empirical evaluation, which uncovers advantages and disadvantages of different types of task reformulations.

Brief summary of the review:
This paper was a pleasure to read. The quality of the work is very high and suitable to be presented at a top AI conference. It is written fluently, the formal presentation is concise and clear, and simple examples guide the reader through the definitions. Moreover, the framework generalizes multiple previously introduced planning task reformulations, and FTS-based planning tasks warrant a deeper look into heuristic search performed on such tasks, therefore this work fits perfectly into the scope of the workshop.

Detailed review:
The paper is very dense, yet it manages to cite relevant related work. One can note though, that while the introduction mentions that reformulations are a common tool to reduce the accidental complexity of a planning task, this is never discussed again later on. I wonder how the presented reformulation methods impact the accidental complexity.

The presentation of FDR/FDS representations, the M&S framework and the various transformations is concise, clear, formally correct, and easy to follow. The plan reconstruction is a bit harder to follow, although being familiar with cascading tables helps here. The example really helps to completely grasp the reconstruction process. What I find a bit odd is that the paper mentions that it avoids pruning-based reformulations because of the overhead, yet Proposition 1 includes pruning-based reformulations. Since the proof considers the previously shown plan reconstruction process I wonder if Proposition 1 really should include pruning (I do not necessarily think it is wrong, as I guess the overhead for pruning does not really affect the poly-time property). I would also have expected a part of the empirical evaluation to discuss this overhead, yet it is not mentioned anywhere. I think the paper would benefit from one or two more sentences in how much overhead one can expect when using pruning (maybe the overhead warrants the pruning power?).

The relation to other FDR reformulation methods is also clearly written and interesting to follow, although one could argue that 'dominating' is not necessarily the most fitting description for a method which can perform the same formulations (why not require the dominating method to provide additional reformulations, both, shrinking and label reduction, would still dominate in this case, no?).

The evaluation is also very comprehensive (it evaluates the different techniques over *all* planning tasks of the previous IPCs) and provides interesting insights. While it mostly covers the results over all domains, I think an analysis on the effect of the different transformations on particular domains could be interesting. While this is partially hinted to in the conclusion, the paper could benefit from a short discussion on which types of transformation might be particular useful (or harmful) in what type of domain.

Minor comments:

Plan reconstruction:
- 'On the final task, we can run any planning algorithm on /Pi_n [...] to find a plan [...] for /Pi_n'. Since the final task is /Pi_n this sentence mentions three times the same task.

Reconstruction of MLB Reformulations:
- /lambda(l) is not defined before or later on. (I suspect /lambda was renamed to f^L?)
- Example: 'keeping track of one of the factored state that corresponds' => one of the factored states

- /tau-label reconstrunction:
 - "to some in \Pi_i" => to some TS in \Pi_i
 - you use /alpha for the weak bisimulation of /Theta, but before /alpha was usually used as the abstraction function for states, and not transition systems.

- Proof of Prop 2: why can computing the coarses weak bisimulation of a TS be done in poly time?
- Def. 4: I suggest to use two different naming/font schemes to differentiate between the reformulation methods of FTS and FDR tasks.
- Figure 2: The figure could explicitly say which plot refers to weak bisimulation shrinking.
- Discussion of Fig. 3:
 - 'This plots' => This plot
 - 'Tt is also remarkable the large amount of instances' => grammar
 - '(there is no expansions until last jump)': first I wanted to mention grammar, but then I remembered that this is one of the search properties for FD benchmarks. I suspect this confuses more people than it helps people familiar with FD benchmarks, so I would suggest to just remove this remark.

---

### Author Response · Authors · 2019-04-11
**Thanks for your reviews**

Thanks to both reviewers for their comments. We will use them to improve the paper. Next, we give some replies:
  * "I wonder how the presented reformulation methods impact the accidental complexity."

   Accidental complexity is caused by the choice on how the problem is encoded and our
   methods are aimed to reformulate this. For example, by merging some variables one can
   effectively affect the causal graph of the planning task (e.g. by removing unreachable
   and irrelevant parts of a transition system).

   In the paper we compared our methods against (Tozicka et al. 2016) which is the more
   recent paper that we could find regarding reformulation/reduction of planning
   tasks. But we will elaborate the relation to the paper by Haslum, 2007, as it is also
   closely related. For example, the abstraction methods of Haslum prove that if the "free
   DTG" of a variable has a single strongly connected component then the variable can be
   safely removed away. This is closely related to our weak-bisimulation shrinking, which
   is also an abstraction method. Indeed, if the condition in (Haslum, 2007) holds, free
   edges in the DTG will be tau-labels in our method and weak-bisimulation will "shrink
   away" the variable too.

   Moreover, in cases where there are no such variables we can still achieve some reductions by taking advantage of one or more of the following:
      * tau-labels are more general than free variables
      * the "bisimulation" aspect of our shrinking
      * some reductions can be achieved even if the variable cannot be completely removed.
      * merge reformulations can help to achieve more tau-labels

   * "What I find a bit odd is that the paper mentions that it avoids pruning-based
     reformulations because of the overhead, yet Proposition 1 includes pruning-based
     reformulations."

     On the contrary, we always apply all pruning methods until a fixpoint is reached
     because they usually have a positive effect and the overhead is not too high.

   Perhaps the sentence that is confusing is "pruning cannot be performed often in the
   atomic representation due to the preprocessing of FD". What we meant is that, as FD
   already removes irrelevant/unreachable facts from the planning task, our pruning
   methods do not often find anything to else to prune (until merge reformulations are
   applied).

  * About why not require "dominance" to mean better than:

   We used the word "dominance" as "at least as good" as it is done, for example for
   heuristics (a heuristic h dominates another h' if h(s) >= h'(s) for all s). If both
   methods dominate each other, it means that they are equivalent.

   * About discussion on which types of transformation might be particular useful (or
     harmful) in what type of domain:

     Yes, we'll add some comments about this. Domains where shrinking is most useful are
     those where entire variables can be abstracted away. Domains where merge
     reformulations are useful involve cases where symmetries (e.g. childsnack) or
     unreachable/irrelevant tuples of facts (e.g. woodworking, tpp) that involve a subset
     of variables and can be discovered after a merge reformulation. However, since merge
     reformulations usually negatively affect delete-relaxation heuristics (as happens
     with other reformulations, e.g. Baggy) the impact on state space size has to be very
     significant to compensate for this.

   * "Binary variables example"
     The states (0,0), (0,1), and (1,0) are indeed bisimilar. The word "counter" was
     missleading in this context, because from (1, 0) there are transitions to (0, 0), (0,
     1), (1, 0) and (1, 1). This is just a very extreme example where all states can be
     reached from any state. Therefore, all states are bisimilar except the goal if we are
     considering a goal-respecting bisimulation.

   * Figure 2"
     The top row is for optimal planning (using bisimulation) and the bottom row for
     satisficing planning (using weak bisimulation.

   * About new domains being solved. Not with shrink/label reduction
     transformations. Actually, we included elevators and transport in the list of domains
     solved by our reformulations, which are not solved by Tozicka et al. This is not
     precise, since many instances of those domains may require some search, but they are
     simplified in a degree that blind search easily solves all instances in the optimal
     benchmark set (the last instance in elevators-opt14 is only solved if merge
     reformulations are enabled).

     Thanks to the merge reformulations, we are able to entirely simplify small tasks in
     some domains too. But this is less significant since it may depend on the relation
     between the bound of abstract states chosen and the size of the state space.

---

> ### Comment · AnonReviewer1 · 2019-04-15
>
> Thank you for the insightful response. This answers most of my questions, two comments:
>
> > We used the word "dominance" as "at least as good" as it is done, for example for heuristics (a heuristic h dominates another h' if h(s) >= h'(s) for all s). If both methods dominate each other, it means that they are equivalent.
>
> I understand that. My point is that both, label reduction and weak bisimulation are more general than the corresponding FDR methods, so they are not equivalent. This is of course subjective, but I think it is more explicit if this is just included in both theorems, instead of described informally afterwards.
>
> >On the contrary, we always apply all pruning methods until a fixpoint is reached because they usually have a positive effect and the overhead is not too high.
> > Perhaps the sentence that is confusing is "pruning cannot be performed often in the atomic representation due to the preprocessing of FD".
>
> The sentence that confuses me is in Section Plan Reconstruction:
>
> > We avoid this by ignoring pruning-based reformulations ([...]) and aggregating sequences of reformulations that correspond to merge, label reduction, and bisimulation transformation.
>
> Could you clarify what you mean by "ignoring pruning-based reformulations"?

---

> > ### Author Response · Authors · 2019-04-15
> > **reply**
> >
> > About dominance:
> > yes, you're right, thanks for the good advice. We'll rephrase the Theorems to include that the dominance is actually strict.
> >
> > About pruning-based reformulations:
> > What we meant is that these reformulations can be ignored by the plan reconstruction procedure. More formally, If T is a task and T' is a reformulated task where the reformulation applied some pruning method. Any plan for T' is directly a plan for T so no reconstruction is necessary. Everything that has been pruned (e.g. fact, label, transition) is simply not present in the reformulated task so it is not part of the solution found. This is because pruning methods can only eliminate paths in the state space, but they otherwise do not modify the existing paths or introduce any new paths that were not present before (as abstraction does, for example)
> >
> > We'll clarify this on the paper.

---

> > > ### Comment · AnonReviewer1 · 2019-04-16
> > > **Thank You**
> > >
> > > Thank you, now that you lay it out it seems obvious.

---

### Meta-Review · Program_Chairs · 2019-04-25

**Recommendation:** Accept
**Confidence:** 5

**Metareview:**

Dear Authors,
thank you very much for your submission. We are happy to inform you that
we have decided to accept it and we look forward to your talk in the workshop.
Please, go over the feedback in the reviews and correct or update your papers
in time for the camera ready date (May 24).
Best regards
HSDIP organizers